# Polar surface structure of oxide nanocrystals revealed with solid-state NMR spectroscopy

Junchao Chen [1,6], Xin-Ping Wu [2,3,6]*, Michael A. Hope [4,6], Kun Qian[1], David M. Halat [4], Tao Liu[4], Yuhong Li[1], Li Shen[1], Xiaokang Ke[1], Yujie Wen [1], Jia-Huan Du [1], Pieter C.M.M. Magusin[4], Subhradip Paul [5], Weiping Ding[1], Xue-Qing Gong[2], Clare P. Grey[4]* & Luming Peng [1]*

Compared to nanomaterials exposing nonpolar facets, polar-faceted nanocrystals often exhibit unexpected and interesting properties. The electrostatic instability arising from the intrinsic dipole moments of polar facets, however, leads to different surface configurations in many cases, making it challenging to extract detailed structural information and develop structure-property relations. The widely used electron microscopy techniques are limited because the volumes sampled may not be representative, and they provide little chemical bonding information with low contrast of light elements. With ceria nanocubes exposing (100) facets as an example, here we show that the polar surface structure of oxide nanocrystals can be investigated by applying $^{17}O$ and $^{1}H$ solid-state NMR spectroscopy and dynamic nuclear polarization, combined with DFT calculations. Both $CeO_4$-termination reconstructions and hydroxyls are present for surface polarity compensation and their concentrations can be quantified. These results open up new possibilities for investigating the structure and properties of oxide nanostructures with polar facets.

---

[1] Key Laboratory of Mesoscopic Chemistry of MOE and Collaborative Innovation Center of Chemistry for Life Sciences, School of Chemistry and Chemical Engineering, Nanjing University, 163 Xianlin Road, Nanjing 210023, China. [2] Key Laboratory for Advanced Materials, Centre for Computational Chemistry and Research Institute of Industrial Catalysis, East China University of Science and Technology, 130 Meilong Road, Shanghai 200237, China. [3] Department of Chemistry, Chemical Theory Center, and Supercomputing Institute, University of Minnesota, 207 Pleasant Street SE, Minneapolis, MN 55455-0431, USA. [4] Department of Chemistry, University of Cambridge, Lensfield Road, Cambridge CB2 1EW, UK. [5] DNP MAS NMR Facility, Sir Peter Mansfield Magnetic Resonance Centre, University of Nottingham, Nottingham NG7 2RD, UK. [6] These authors contributed equally: Junchao Chen, Xin-Ping Wu, Michael A. Hope. *email: xpwu@ecust.edu.cn; cpg27@cam.ac.uk; luming@nju.edu.cn

Polar surfaces, which have a permanent dipole moment perpendicular to the surface, are of great importance in both physical and chemical applications[1–4]. Due to the very large energies of uncompensated surfaces, polarity compensation is required, generating different and complex surface configurations for these facets. Therefore, it is extremely difficult to understand the atomic-scale structure of polar surfaces, which is essential in order to design related nanomaterials for a targeted property[5–8]. By applying electron microscopy and computational modelling, a variety of polarity compensation mechanisms have been proposed, including ordered surface reconstructions[9], surface disorder[10], adsorption of environmental gas molecules[11], surface metal layers deposition[12], and subsurface oxygen vacancies[13]. Despite the many advantages of microscopy techniques, they are limited to the visualization of a small fraction of the sample which may not yield reliable quantitative information about the whole sample, and are typically performed at high vacuum conditions that may alter the surface environment[14]. Furthermore, light elements, such as hydrogen and oxygen which are of key importance for many materials, are difficult to probe with such techniques[15]. Although significant developments have been made in environmental electron microscopy, which allows materials to be investigated under adjustable pressure conditions and in variable gaseous environments[16–18], the other disadvantages remain and complementary methods are required.

Solid-state NMR spectroscopy is a powerful method which can provide rich local structural information for solids[19–26], complementary to the information obtainable from diffraction[27] and microscopy techniques. Recently, $^{17}O$ solid-state NMR spectroscopy has been developed as a new approach for determining the surface structure of oxide nanomaterials, with help from surface-selective labeling and DFT calculations. Oxygen ions in different layers of ceria nanostructures[28] and at different facets of anatase titania nanocrystals[29] can be distinguished according to the NMR shifts. However, only non-polar facets were studied previously and no attempt was made to investigate the more challenging oxide nanostructures with polar facets. Furthermore, quantification of different surface species was not possible using exclusively $^{17}O$ NMR, due to the quadrupolar nature of $^{17}O$ and the potentially non-uniform isotopic labeling procedure.

Ceria nanocubes expose (100) facets, which show exceptional properties as both the catalytically active plane and the support facet; this is a relatively simple polar surface, making ceria nanocubes an ideal model[30–33]. Using the example of ceria nanocubes, we introduce a strategy of qualitative $^{17}O$ and quantitative $^{1}H$ solid-state NMR spectroscopy combined with DFT calculations to characterize oxide nanocrystals with polar facets. We thereby quantitatively determine detailed polar surface structural information, specifically the presence and concentration of reconstructed Ce terminated structures (CeO$_4$-t) and hydroxyl groups.

## Results

**Morphology of the ceria nanocubes.** Ceria nanocubes were hydrothermally synthesized with Ce(NO$_3$)$_3$·6H$_2$O and NaOH (see methods). The X-ray diffraction (XRD) data (Supplementary Fig. 1) confirms the formation of ceria with a fluorite structure (JCPDS No. 34-0394). High-resolution transmission electron microscopy (HRTEM) images show that the samples adopt a cubic morphology with sizes of 18 to 40 nm, dominated by (100) polar surfaces before and after $^{17}O$ enrichment (Supplementary Fig. 2). Inductively coupled plasma mass spectrometry (ICP-MS), elemental analysis and X-ray photoelectron spectroscopy (XPS) data show that there are no detectable Na$^+$ or NO$_3^-$ impurities (Supplementary Fig. 3 and Supplementary Table 1). In addition,

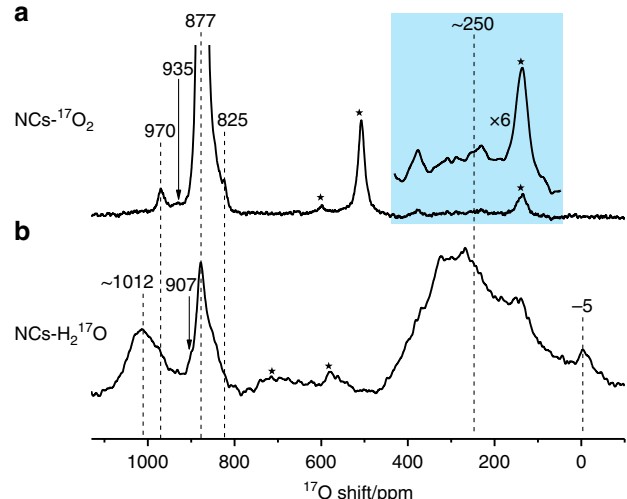

**Fig. 1** $^{17}O$ Solid-State NMR spectra of ceria nanocubes. The $^{17}O$ NMR measurements were performed at a spinning speed of 20 kHz for NCs-$^{17}O_2$ (**a**) and 16 kHz for NCs-H$_2$$^{17}O$ (**b**). A rotor synchronized Hahn-echo pulse sequence ($\pi/6$-$\tau$-$\pi/3$-$\tau$-acquisition) with $^1H$ decoupling and a 0.5 s recycle delay was used. Stars denote spinning sidebands. In order to show the peaks due to OH better, a line broadening of 600 Hz is applied for the enlarged spectrum shown in light blue region.

the concentration of oxygen vacancies at the surface is only 1.0% according to analysis of Raman spectroscopy data and, therefore, their influences are not considered further (Supplementary Fig. 4 and Supplementary Note 1).

**NMR spectra of the ceria nanocrystals.** The $^{17}O$ solid-state NMR spectra of ceria nanocubes enriched with $^{17}O_2$ at 523 K (NCs-$^{17}O_2$) and H$_2$$^{17}O$ at 373 K (NCs-H$_2$$^{17}O$) are shown in Fig. 1. The $^{17}O$ NMR spectrum of NCs-$^{17}O_2$ is dominated by the peak at 877 ppm (see the untruncated spectrum in Supplementary Fig. 5) due to the OCe$_4$ environment in the bulk of the ceria nanocubes, but other signals can also be observed at 970, 935, and 825 ppm. $^{17}O$ NMR signals for ceria samples with (111) facets have previously been observed at 1040, 920, and 825 ppm due to oxygen ions in the first, second, and third (sub-)surface layers respectively[28]; the shoulder resonance at 825 ppm in the NCs-$^{17}O_2$ spectrum is therefore most likely due to a deeper sub-surface layer while the signals at 970 and 935 ppm, which have not previously been observed, may be tentatively assigned to the oxygen ions at the (100) surface.

For the spectrum of NCs-H$_2$$^{17}O$, in addition to the bulk signal with a maximum at 877 ppm, two broad peaks centered at approximately 250 and 1012 ppm can be observed. The former is most likely to be related to surface hydroxyl groups (Ce-OH)[29,34], and on closer inspection can also just be distinguished for NCs-$^{17}O_2$. The latter can again, based on its high frequency compared to the bulk resonance, be attributed to under-coordinated surface oxygen species. Surface-selective isotopic labeling is achieved by exposing the samples to $^{17}O_2$ gas or to H$_2$$^{17}O$ vapor at relatively low temperatures, although there are differences between the spectra that will be discussed later.

A possible explanation for the higher frequency signal in the spectrum of NCs-H$_2$$^{17}O$ is the formation of some degree of the thermodynamic (111) surface, given that the first surface layer in this case has been observed at 1040 ppm. However, the HRTEM images show little evidence for (111) facets (Supplementary Fig. 2), and after re-enriching NCs-H$_2$$^{17}O$ with $^{17}O_2$ gas, the surface signals are identical to those of NCs-$^{17}O_2$ and do not

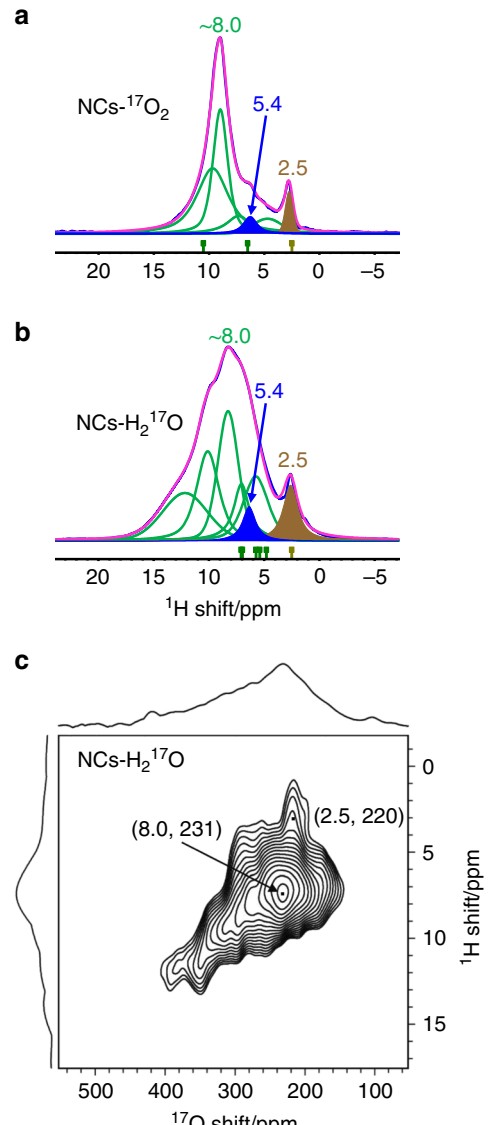

**Fig. 2** $^1$H Solid-State NMR spectra and $^1$H → $^{17}$O HETCOR NMR of ceria nanocubes. The $^1$H NMR measurements for NCs-$^{17}$O$_2$ (**a**) and NCs-H$_2$$^{17}$O (**b**) were performed at a spinning speed of 12 kHz using a rotor synchronized Hahn-echo pulse sequence (π/2-τ-π-τ-acquisition) with a recycle delay of 2.0 s. Deconvolution of $^1$H spectra and the summarized $^1$H chemical shifts (obtained from DFT calculations, see below) are also shown. Background signals from the cap, rotor and probe etc. are removed by subtracting the spectra of an empty rotor obtained at the same condition from the initial spectrum. **c** The $^1$H → $^{17}$O HETCOR spectrum of NCs-H$_2$$^{17}$O, recorded at a MAS rate of 14 kHz with a recycle delay of 2 s and a contact time of 0.5 ms.

summary of these deconvoluted signals is presented in Supplementary Table 2. The signal at 5.4 ppm is most likely due to molecularly adsorbed water molecules and the 2.5 ppm signal can be assigned to terminal hydroxyl species (-OH$_T$)[38]; the remaining intensity is then assigned to bridging hydroxyl groups (-OH$_B$) with a distribution of environments and degrees of hydrogen bonding. A similar $^1$H NMR spectrum was observed for NCs-H$_2$$^{17}$O (Fig. 2b), but with a broader signal centered around 8 ppm, corresponding to bridging hydroxyl groups with an even greater distribution of environments and hydrogen bonding. Quantitative analysis of the integrated $^1$H intensities, combined with the specific surface areas as measured from the BET isotherms, yields adsorbed water contents for NCs-$^{17}$O$_2$ and NCs-H$_2$$^{17}$O of 2.6 and 3.7 H$_2$O molecules per surface unit (59.3 Å$^2$), respectively. Of these, 0.2 H$_2$O molecules are molecularly adsorbed for both NCs-$^{17}$O$_2$ and NCs-H$_2$$^{17}$O, while the rest are dissociatively adsorbed to form hydroxyl groups (see Supplementary Fig. 7 and Supplementary Table 3). The quantitative $^1$H NMR spectroscopy indicates that there are similar concentrations of hydroxyl groups on both NCs-$^{17}$O$_2$ and NCs-H$_2$$^{17}$O, however the intensity of signals from hydroxyl groups in the $^{17}$O NMR spectrum of the former is much lower. This is ascribed to unenrichment of the oxygen atoms of the hydroxyl groups for NCs-$^{17}$O$_2$ by natural abundance water, which is not completely dehydrated at the relatively low temperature of 523 K. In contrast, the surface water of NCs-H$_2$$^{17}$O is replaced by H$_2$$^{17}$O, so the same unenrichment does not occur.

The $^1$H → $^{17}$O HETCOR NMR spectrum of NCs-H$_2$$^{17}$O (Fig. 2c) shows correlations between the $^{17}$O signals at around 250 ppm with the $^1$H signals due to hydroxyl groups, confirming the assignment of the $^{17}$O NMR spectra. Furthermore, hydroxyl sites with higher $^{17}$O shifts are associated with higher $^1$H shifts, and thus a stronger acidity (although stronger hydrogen bonding can also result in larger shifts)[39]. The conclusion that bridging hydroxyl groups are more acidic than terminal hydroxyls is in agreement with previous reports in zeolites[39]. Since the $^{17}$O NMR shift range is much wider than for $^1$H NMR shifts, $^{17}$O NMR spectroscopy may provide an alternative and more sensitive probe of the acidity and acid-catalysis reactivity for oxide nanomaterials. However, $^{17}$O NMR experiments often require spectra acquired at multiple magnetic fields, preferably higher fields, or with high resolution techniques (e.g., MQMAS[40]), in order to decrease the linewidths arising from quadrupolar interactions.

**Spectral Assignments from DFT Calculations.** DFT calculations have previously proved successful in aiding spectral assignment for surface oxygen ions in oxide nanostructures[28,29]. The differences between the calculated and experimental results are generally around or less than 10 ppm[28], allowing reliable spectral assignment. An oxygen terminated (O-t) model of ceria (100) surface was previously investigated for DFT calculations[28] (Fig. 3a), however, the calculated $^{17}$O NMR parameters (Supplementary Figs. 8–10 and Supplementary Table 4) are not in agreement with our NMR observations, i.e., no surface species in the calculations are associated with resonant frequencies at 970, 935 or ~1012 ppm and the calculated signal for the 1$^{st}$ layer two-coordinate oxygen ion (O$_{2C}$), with a high frequency chemical shift (1117 ppm), is not observed in the experimental spectrum.

As seen from the $^1$H and $^{17}$O NMR spectra, a significant number of hydroxyl groups are present at the surface, which must be considered. A previous computational study reported that dissociative adsorption of water is much more favorable than the molecular adsorption of water on the (100) O-t surface[41]; accordingly, DFT calculations were performed on O-t models with 1, 2, 3 and 4 dissociatively adsorbed H$_2$O molecules per O-t

exhibit signals associated with (111) surfaces (Supplementary Fig. 6). This suggests that the high frequency signal is related to the (100) surface itself and the H$_2$$^{17}$O enrichment.

To investigate the presence of hydroxyl groups or molecularly adsorbed H$_2$O on the ceria surface, quantitative $^1$H NMR was performed. $^1$H chemical shifts are sensitive to hydrogen bonding[35,36] and the lineshapes can be affected by the distribution of hydrogen bond distances; in particular, stronger hydrogen bonding results in higher frequency $^1$H NMR signals[37]. The spectrum of NCs-$^{17}$O$_2$ in Fig. 2a shows a broad resonance between 2–16 ppm (the sum of the green deconvoluted Lorentzian functions), along with a shoulder at 5.4 ppm (blue signal), and a relatively sharp peak at 2.5 ppm (brown signal); a

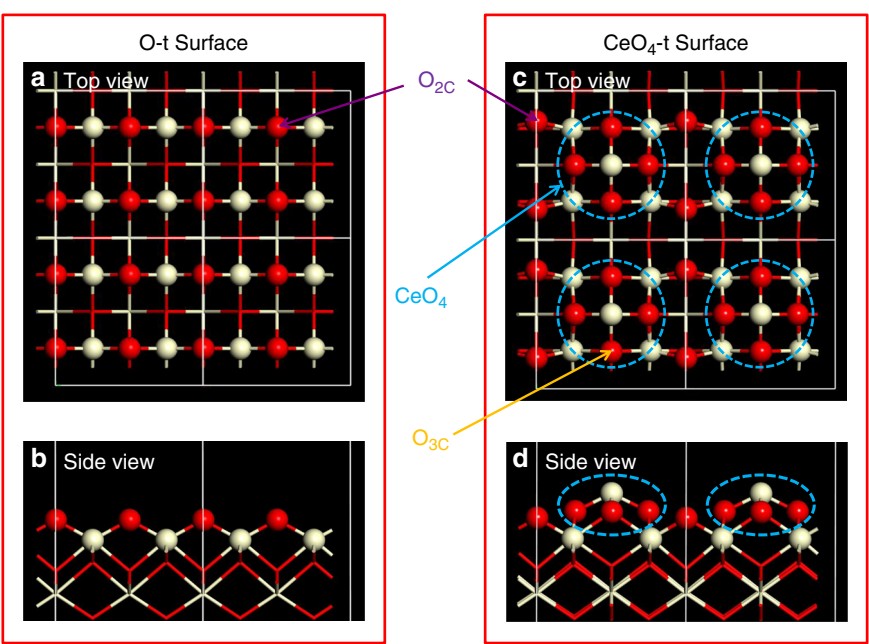

**Fig. 3** Structure of O-t and CeO$_4$-t surface. Surface oxygen ions are represented as red balls and cerium ions as off-white, in the top and side views of ceria (100) facets with an O-t surface (**a**, **b**) and a CeO$_4$-t surface (**c**, **d**). The top views (**a**, **c**) each show four surface units (each surface unit is 7.7 Å × 7.7 Å = 59.3 Å$^2$), delimited by white boxes. Each O-t surface unit includes four O$_{2C}$ ions, while each CeO$_4$-t surface unit contains two O$_{2C}$ ions and one CeO$_4$ reconstruction (blue dashed circle), with four three-coordinated oxygen ions (O$_{3C}$).

surface unit (Supplementary Figs. 11–23 and Supplementary Tables 5–8). On reaction with water, the under-coordinated O$_{2C}$ sites with high chemical shifts are protonated to form hydroxyl groups with calculated shifts of 191–360 ppm, as observed in the experimental $^{17}$O NMR spectra; four dissociated H$_2$O molecules per O-t surface unit are required to protonate all the O$_{2C}$ sites (monolayer hydroxylation, see Supplementary Fig. 20). However, there are no calculated resonances at 970, 935 or ~1012 ppm; this suggests that the structure of polar (100) facets in ceria nanocubes is more complicated than the simple O-t model.

Two recent studies suggested that a fraction of ceria (100) facets may form CeO$_4$ terminated (CeO$_4$-t) reconstructions, which yield a lower surface energy than cerium terminated (Ce-t) or O-t surfaces[10,42]. Therefore, DFT calculations were performed on a model comprising CeO$_4$-t reconstructions linked by O$_{2C}$ sites (CeO$_4$-t surface, Fig. 3b)—a pure CeO$_4$-t reconstruction has previously been shown to have a high surface energy and is thus unstable[10].

First, the relative energies of dissociative and molecular adsorption were calculated for a single H$_2$O molecule per CeO$_4$-t surface unit (Supplementary Figs. 24–26). A comparison of the adsorption energies shows that H$_2$O molecules also prefer to adsorb dissociatively on clean CeO$_4$-t ceria (100) surfaces (Supplementary Table 9). The models M$_0$, M$_1$, M$_2$ and M$_3$ then correspond to 0, 1, 2 and 3 dissociatively adsorbed H$_2$O molecules per CeO$_4$-t surface unit. The most energetically favorable H$_2$O adsorption sites in M$_1$, M$_2$, and M$_3$ were then determined by calculating and comparing the adsorption energies of several possible structural models (the lowest energy structure for each model is shown in Supplementary Figs. 26–28). The corresponding $^{17}$O NMR calculation results for M$_0$, M$_1$, M$_2$, and M$_3$ are shown in Figs. 4 and 5, Supplementary Figs. 29–38 and Supplementary Tables 10–13 (also see more discussion in Supplementary Note 2 and Supplementary Fig. 39).

The O$_{2C}$ ions in the first layer of models M$_0$ and M$_1$ are associated with very high chemical shifts of 1162 and 1168 ppm respectively; such high frequency signals are not present in the

experimental $^{17}$O NMR spectra. This again indicates that these high energy, under-coordinated species have reacted with water to form more stable hydroxyl species, as observed in the $^1$H and $^{17}$O NMR spectra.

The calculated $^{17}$O NMR shifts (center of gravity, $\delta_{CG}$) based on the CeO$_4$-t model with two or three H$_2$O molecules dissociatively adsorbed on each surface unit (M$_2$ and M$_3$, Figs. 4 and 5, Supplementary Figs. 35–38, Supplementary Tables 12 and 13) are in good agreement with the experimental spectra. All the O$_{2C}$ ions on the surface are consumed by reacting with H$_2$O molecules to form hydroxyl groups, therefore there is no calculated signal at a shift higher than 1100 ppm, as observed experimentally. For the O$_{3C}$ surface species in the CeO$_4$ reconstruction, the calculated shifts in M$_2$ are 943 and 968 ppm, which match the observed resonances at 935 and 970 ppm in the experimental spectrum of NCs-$^{17}$O$_2$, while the calculated O$_{3C}$ shifts in M$_3$ are 910 and 996 ppm, in accordance with the experimental signals at 907 and ~1012 ppm for NCs-H$_2$$^{17}$O. The resonance at ~1012 ppm is quite broad, which is presumably due to a distribution of local environments caused by different arrangements of nearby dissociative H$_2$O molecules. The observation that the resonant frequencies are higher for lower-coordinated oxygen ions on the surface is similar to the previous studies on ceria nanoparticles exposing (111) facets and titania nanostructures[29,34].

The DFT calculations predict that the O$_{4C}$ ions in the 2$^{nd}$ or 3$^{rd}$ layer for both NCs-$^{17}$O$_2$ and NCs-H$_2$$^{17}$O have resonant frequencies closer to the bulk shift of 877 ppm, which is consistent with the relatively broad component observed experimentally for the peak centered at 877 ppm. The predicted peak in the 2$^{nd}$ layer at 756 ppm for M$_2$ is not experimentally observed; this environment, although fully coordinated, is highly distorted (see Supplementary Fig. 40). This distortion may be lost on addition of a bridging hydroxyl between the CeO$_4$ reconstructions (e.g. H$_4$, see Fig. 6), which is above the distorted environment, or the distortion may be averaged at non-zero temperatures due to rapid interconversion of different local environments with similar energies.

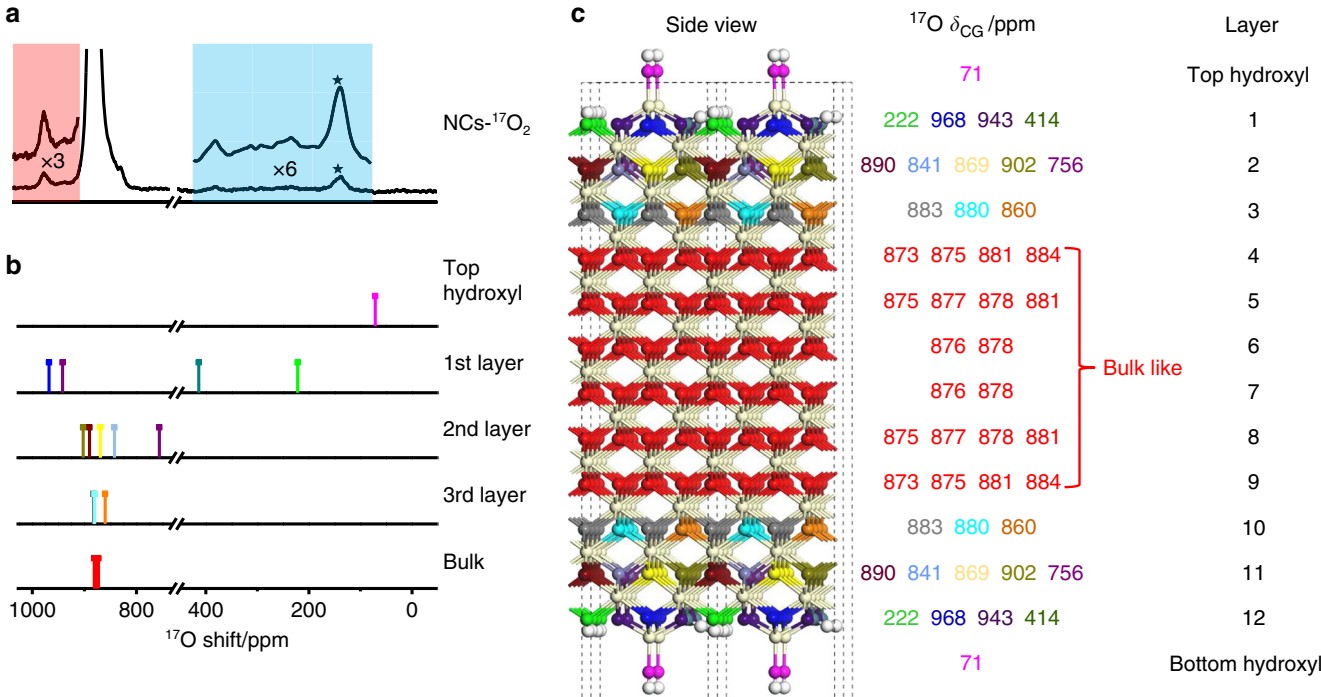

**Fig. 4** $^{17}O$ NMR spectrum, calculated $^{17}O$ NMR shifts and the structure model of ceria NCs-$^{17}O_2$. **a** $^{17}O$ Solid-State NMR spectra of ceria NCs-$^{17}O_2$ (shown in Fig. 1). **b** The summarized $^{17}O$ NMR shifts ($\delta_{CG}$s) predicted for the CeO$_4$-t model with two H$_2$O molecules dissociatively adsorbed on each surface unit (M$_2$). **c** The hydrated CeO$_4$-t model used in the DFT calculations with NMR shifts ($\delta_{CG}$s) for oxygen ions and the layer number shown on the right. Red, off-white, and white balls represent bulk oxygen, cerium, and hydrogen ions, respectively. Surface oxygen ions with different shifts are shown in different colors. The calculated $^{17}O$ NMR parameters for each oxygen ion are shown in Supplementary Table 12. A line broadening of 600 was applied to the enlarged spectrum in the blue region, no line broadening was applied to the enlarged spectrum in the red region.

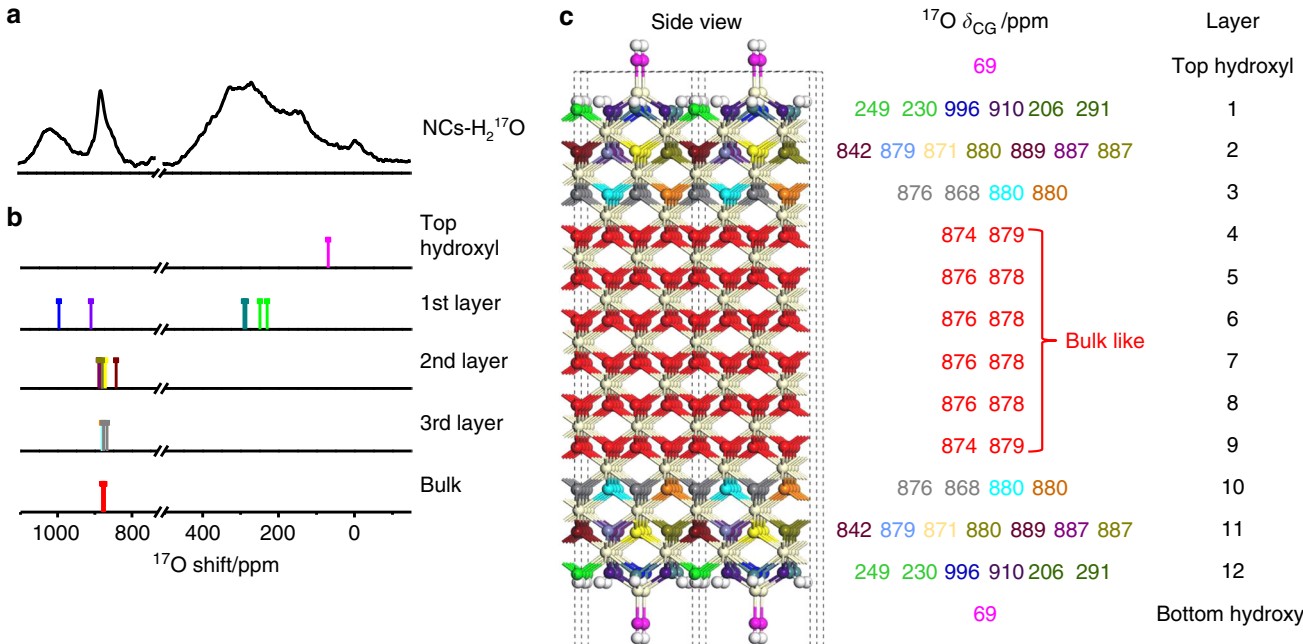

**Fig. 5** $^{17}O$ NMR spectrum, calculated $^{17}O$ NMR shifts and the structure model of ceria NCs-H$_2$$^{17}O$. **a** $^{17}O$ Solid-State NMR spectra of ceria NCs-H$_2$$^{17}O$ (shown in Fig. 1). **b** The summarized $^{17}O$ NMR shifts ($\delta_{CG}$s) predicted for the CeO$_4$-t model with three H$_2$O molecules dissociatively adsorbed on each surface unit (M$_3$). **c** The hydrated CeO$_4$-t model used in the DFT calculations with calculated NMR shifts ($\delta_{CG}$s) for oxygen ions and the layer number shown on the right. Red, off-white, and white balls represent bulk oxygen, cerium, and hydrogen ions, respectively. Surface oxygen ions with different shifts are shown in different colors. The calculated $^{17}O$ NMR parameters for each oxygen ion are shown in Supplementary Table 13.

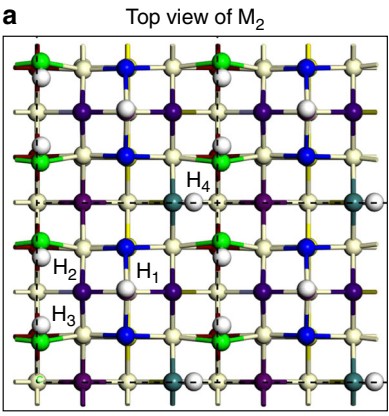

Top view of M$_2$

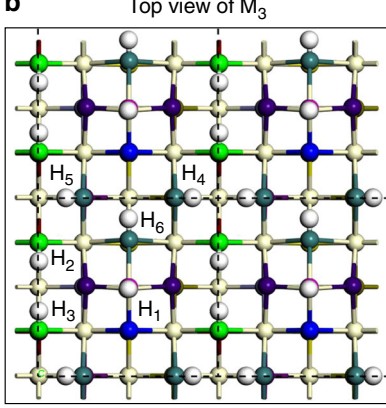

Top view of M$_3$

**Fig. 6** Structure of M$_2$ and M$_3$. White and off-white spheres represent hydrogen and cerium ions, respectively, in the top view of M$_2$ (**a**) and M$_3$ (**b**). Surface oxygen groups with different chemical shifts are in different colors. For each CeO$_4$-t surface unit of M$_2$, four hydrogen ions labeled as H$_1$, H$_2$, H$_3$, and H$_4$ form from the two dissociative H$_2$O molecules while for each CeO$_4$-t surface unit of M$_3$, six hydrogen ions labeled as H$_1$, H$_2$, H$_3$, H$_4$, H$_5$, and H$_6$ form from the three dissociative H$_2$O molecules.

**Table 1 Calculated $^1$H NMR chemical shifts$^a$ for M$_2$ and M$_3$. The corresponding structures are presented in Fig. 6.**

| H$_{ion}$ No. | M$_2$ | | M$_3$ | |
|---|---|---|---|---|
| | $\delta_{iso}$/ppm | Assignment | $\delta_{iso}$/ppm | Assignment |
| 1 | 2.50 | –OH$_T$ | 2.50 | –OH$_T$ |
| 2 | 5.56 | –OH$_B$ | 5.72 | –OH$_B$ |
| 3 | 5.70 | –OH$_B$ | 4.79 | –OH$_B$ |
| 4 | 10.18 | –OH$_B$ | 7.07 | –OH$_B$ |
| 5 | — | — | 6.95 | –OH$_B$ |
| 6 | — | — | 5.38 | –OH$_B$ |

$^a$The chemical shifts ($\delta_{iso}$) are referenced to the experimental chemical shift of the terminal hydroxyl (2.50 ppm) (Supplementary Figs. 41 and 42) and summarized under the $^1$H spectra in Fig. 2a, b

**Table 2 The contents of terminal hydroxyls (-OH$_T$ and -OH$_B$) and different surface units.**

| Sample | Hydroxyl$^a$/% | | Surface Unit/% | |
|---|---|---|---|---|
| | -OH$_T$ | -OH$_B$ | O-t | CeO$_4$-t |
| NCs-$^{17}$O$_2$ | 6.3 | 86.1 | 57.1 | 42.9 |
| NCs-H$_2$$^{17}$O | 5.6 | 88.6 | 57.5 | 42.5 |

$^a$The –OH$_T$ and –OH$_B$ contents are determined from the deconvoluted peak intensities of the $^1$H NMR spectra (Fig. 2, Supplementary Table 2)

Terminal hydroxyl sites (-OH$_T$) are calculated to have $^{17}$O shifts ($\delta_{CG}$s) of 69 ppm for M$_2$ and 71 ppm for M$_3$, while bridging hydroxyl sites (–OH$_B$) for both samples have calculated $^{17}$O shifts ($\delta_{CG}$s) of 206–291 ppm. These sites are predicted to have relatively large $C_Q$s (5.5–7.6 MHz), thus the corresponding resonances are expected to exhibit significant second order quadrupolar broadening. This agrees well with the broad resonance observed at lower shifts in the experimental spectrum of NCs-H$_2$$^{17}$O; the hydroxyl groups have been largely unenriched for NCs-$^{17}$O$_2$, see above.

The above results indicate that the concentration of surface hydroxyl groups has a great impact on the NMR shifts of surface oxygen ions. Furthermore, in order to reproduce the experimental $^{17}$O NMR spectra, both dissociated water and CeO$_4$-t reconstructions must be included in the calculations. Thus, the ceria (100) surface can be regarded as a combination of CeO$_4$-t and O-t surface units, where the under-coordinated O$_{2C}$ ions have been converted to bridging hydroxyl groups and terminal hydroxyl groups have formed on some under-coordinated cerium ions.

The $^1$H NMR chemical shifts were also calculated using DFT, confirming the assignment of the $^1$H NMR signals at 2–16 ppm and 2.5 ppm to bridging (–OH$_B$) and terminal (–OH$_T$) hydroxyl groups on the surface (Fig. 6 and Table 1)[38,39]. The calculated $^1$H shift for -OH$_T$ is the most negative, and since the lowest frequency signal in the experimental $^1$H NMR spectrum is the relatively sharp peak at 2.5 ppm, this resonance is assigned to

-OH$_T$. The broad signal is attributed to –OH$_B$. Different –OH$_B$ environments are associated with a range of chemical shifts and the distribution is wider for the model with three dissociatively adsorbed H$_2$O molecules (M$_3$) than that for two (M$_2$); in particular, the very large shift of H$_4$ in M$_2$ is due to hydrogen bonding to the oxygen ions of H$_2$ and H$_3$. This at least partially explains why the experimental -OH$_B$ resonance is broad and why the spectral line width for NCs-H$_2$$^{17}$O is broader than for NCs-$^{17}$O$_2$, given the higher hydroxyl content of the former. An inhomogeneous distribution of dissociated water and variable hydrogen bonding may also contribute to the broadness of the signals.

Based on the quantitative $^1$H NMR data and the above assignments from the DFT calculations, the fractions of CeO$_4$-t and O-t surface units comprising the (100) facets of ceria nanocubes can be determined (Table 2). This is based on the fact that each CeO$_4$-t surface unit contains one characteristic terminal hydroxyl group (–OH$_T$) and either three (M$_2$, Fig. 2a) or five (M$_3$, Fig. 2b) bridging hydroxyl groups (–OH$_B$), while each hydroxylated O-t surface unit contains eight bridging hydroxyls (-OH$_B$) (Supplementary Fig. 21). The fractions of CeO$_4$-t and O-t surface units calculated for NCs-$^{17}$O$_2$ (57.1% for O-t and 42.9% for CeO$_4$-t) are very close to those for NCs-H$_2$$^{17}$O (57.5% for O-t and 42.5% for CeO$_4$-t), supporting the assignment of these models.

**$^{17}$O DNP NMR spectroscopy.** Recent developments in dynamic nuclear polarization (DNP) provide new opportunities to characterize the surface structure of solid materials[34,43,44]. Direct DNP involves transferring polarization from unpaired electrons directly to the nucleus of interest, with the unpaired electrons typically being added in the form of organic biradicals; because the biradicals are external to the particles, and the hyperpolarization mechanism has a 1/r$^6$ distance dependence, the surface can be selectively hyperpolarized and hence observed in the NMR spectrum (surface enhanced NMR spectroscopy, SENS). Indirect DNP, where $^1$H nuclei are first hyperpolarized before

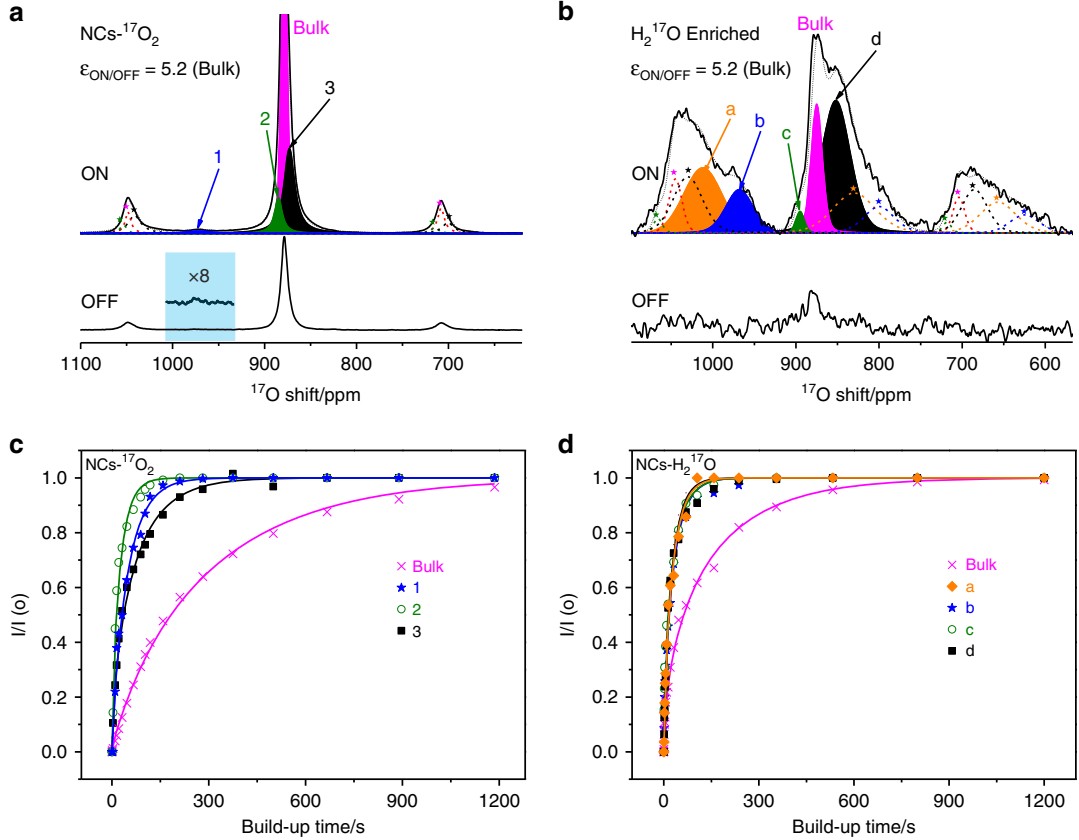

**Fig. 7** $^{17}O$ DNP NMR spectra and saturation recovery data of NCs-$^{17}O_2$ and NCs-H$_2$$^{17}O$. Deconvoluted $^{17}O$ direct DNP NMR spectra, with and without microwave irradiation of **a** NCs-$^{17}O_2$ and **b** NCs-H$_2$$^{17}O$ mixed with the TEKPol radical in dried TCE. The spectra were measured at 100 K at a spinning speed of 13.9 kHz. A pre-saturated rotor synchronized Hahn-echo pulse sequence ($\pi$/2-$\tau$-$\pi$-$\tau$-acquisition) and a recycle delay of 60 s were utilized. Stars denote spinning sidebands. **c**, **d** The corresponding $^{17}O$ direct DNP saturation recovery build-up curves of the deconvoluted isotropic resonances in **a** and **b**. The build-up time ($T_{DNP}$) was determined by deconvoluting the spectrum and fitting the peak areas to a stretched exponential function of the form $\frac{I(t)}{I_0} = 1 - e^{-\left(\frac{t}{T_{DNP}}\right)^{\beta}}$, where $I(t)$ and $I_0$ are the signal intensities at delay $t$ and at equilibrium, respectively, and $\beta$ is the stretching exponent ($0 < \beta < 1$) (Table 3).

cross-polarization to the nucleus of interest, can yield greater enhancements in some cases, but for $^{17}O$ NMR only oxygen atoms directly bonded to H can be observed, so indirect DNP cannot be used to detect the unhydroxylated (sub-)surface oxygen environments[34]. Therefore, direct DNP SENS was applied to study ceria nanocubes and aid spectral assignments. The intensities for the bulk $^{17}O$ peak at 875 ppm are 5.2 times stronger with microwave irradiation than without, for both NCs-$^{17}O_2$ and NCs-H$_2$$^{17}O$ (Fig. 7a, b). For NCs-$^{17}O_2$, the enhancement factor for the peak at 970 ppm is ~8, while the surface signals for NCs-H$_2$$^{17}O$ can only be observed in the "on" spectrum, indicating that hyperpolarization is more efficient for these species than for the bulk, and therefore that these resonances are indeed surface oxygen species.

The DNP build-up time, $T_{DNP}$, can also be fitted to distinguish external and internal $^{17}O$ nuclei[34], since nuclei close to the surface hyperpolarize faster and hence have a shorter $T_{DNP}$. For NCs-$^{17}O_2$, the $T_{DNP}$ values for the peaks at 970, 880, and 870 ppm are much smaller than for the bulk signal at 875 ppm, implying that the former arise from surface species in NCs-$^{17}O_2$ (Fig. 7c and Table 3). Broader peaks are observed for the $^{17}O$ DNP SENS spectra than for the room temperature $^{17}O$ spectra, which can be attributed to the freezing out of motional averaging of the dynamic surface sites at the low temperatures required for DNP, as previously observed for ceria (111) facets[34].

**Table 3 The build-up time ($T_{DNP}$) and stretching exponent ($\beta$) of different peaks extracted from the $^{17}O$ direct DNP saturation recovery build-up curves shown in Fig. 7.**

|  | NCs-$^{17}O_2$ |  |  |  | NCs-H$_2$$^{17}O$ |  |  |  |  |
|---|---|---|---|---|---|---|---|---|---|
|  | Bulk | 1 | 2 | 3 | Bulk | a | b | c | d |
| $\delta_{iso}$/ppm | 875 | 970 | 880 | 870 | 875 | 1012 | 970 | 895 | 853 |
| $T_{DNP}$/s | 329 | 53 | 30 | 95 | 212 | 23 | 38 | 40 | 31 |
| $\beta$ | 0.8 | 0.8 | 0.7 | 0.6 | 0.5 | 0.8 | 0.7 | 0.6 | 0.7 |

The $^{17}O$ DNP spectrum of NCs-H$_2$$^{17}O$ has spinning sidebands which overlap with other resonances due to the lower spinning speeds achievable at 100 K and the higher field at which the DNP experiments were performed (which results in a smaller separation of sidebands in ppm, for the same spinning frequency). Therefore, $^{17}O$ DNP projection magic angle turning and phase adjusted sideband separation (MATPASS) NMR experiments were performed to resolve the isotropic resonances (Supplementary Fig. 43)[45,46]. Four non-bulk resonances at 1012, 970, 895, and 853 ppm can be deconvoluted, and were used to fit the saturation recovery data and obtain the $T_{DNP}$s (Fig. 7d and Table 3). Again, these peaks are associated with shorter $T_{DNP}$s than the peak at 875 ppm arising from the bulk part of the

sample, in agreement with our assignments from conventional $^{17}$O NMR spectroscopy.

## Discussion

$^{17}$O and $^{1}$H solid-state NMR spectroscopy, combined with DFT calculations, were employed to determine the surface structure of ceria (100) polar surfaces. The results obtained in this work provide compelling evidence that $CeO_4$-t reconstructions and hydroxyl groups are present on the surface of ceria nanocubes, both of which are expected to reduce the surface energy and afford polarity compensation. The amount of hydroxyl groups alters the $^{17}$O NMR shifts of oxygen ions at the surface of the nanocubes, making $^{17}$O a very sensitive probe for the surface structure and, therefore, its properties. Furthermore, the fractions of $CeO_4$-t and O-t surface units can be determined with quantitative $^{1}$H NMR measurements. DNP SENS results confirm the $^{17}$O spectral assignments, however, although DNP SENS spectroscopy generally provides a stronger signal-to-noise ratio compared to conventional NMR spectroscopy, due to the restricted spinning rates and broader surface signals at low temperature, certain species can be resolved better with the latter. The strategy introduced here can be applied to gain insight into the surface structures of oxide nanocrystals and materials with polar surfaces.

## Methods

**Preparation of ceria nanocubes.** In a typical synthesis procedure[47], 1.96 g Ce $(NO_3)_3 \cdot 6H_2O$ was added into 40 mL distilled water. After stirring for 5 min, 30 mL NaOH solution (pH = 14) was slowly added into the mixture before it was vigorously stirred for another 30 min at room temperature. The mixture was then transferred into a 100 mL Teflon-lined hydrothermal reactor and heated at 453 K for 24 h before it was allowed to cool to room temperature. The resulting white sediment was centrifuged, washed with distilled water and dried at 353 K overnight. Finally, the solid was annealed in a tube furnace at 573 K for 5 h in flowing oxygen gas to obtain calcined ceria nanocubes.

**Characterization.** Powder X-Ray Diffraction (XRD) characterization was performed with a Philips X'Pro X-ray diffractometer with Cu Kα irradiation (λ = 1.54184 Å) operating at 40 kV and 40 mA. High-Resolution Transmission Electron Microscope (HRTEM) images were recorded on a JEOL JEM-2010 instrument at an acceleration voltage of 200 kV. X-ray Photoelecton Spectra (XPS) were measured on a Thermo ESCALAB 250 X with Al Kα (hv = 1486.6 eV) as the excitation source. The binding energies in XPS spectra were referenced to C 1 s = 284.8 eV. Brunauer–Emmett–Teller (BET) specific surface area information was obtained from nitrogen adsorption at 77 K on a Micromeritics ASAP 2020 system. Raman spectra were acquired with a Bruker Multi RAM FT-Raman spectrometer using 514 nm light from a He–Ne laser source. The content of Na ions was analyzed by an Optima 5300DV inductively coupled plasma mass spectrometer (ICP-MS) while the N content was determined with a Heraeus CHN-0-Rapid elemental analyzer.

**$^{17}$O Isotopic labeling procedure.** Ceria nanocubes were $^{17}$O-labeled on a vacuum line with commercial 90% $^{17}$O-enriched $O_2$ gas and $H_2O$, respectively (Isotec Inc.). In a typical $^{17}$O isotopic labeling procedure, 200 mg sample was placed in a glass tube and calcined at 523 K under $1 \times 10^{-3}$ Torr for 3 h, in order to remove most of the physically adsorbed water and surface hydroxyl groups. The nanocubes were allowed to cool down to room temperature before $^{17}O_2$ gas or $H_2^{17}O$ vapor was introduced into the glass tube. The glass tube was sealed and then heated to the target temperature (523 K for $^{17}O_2$ and 373 K for $H_2^{17}O$ labeling) for 10 h. After the enrichment with $H_2^{17}O$, the samples were exposed to vacuum to remove physisorbed water.

**Solid-State NMR Spectroscopy.** $^{17}$O and $^{1}$H Magic Angle Spinning Nuclear Magnetic Resonance (MAS NMR) experiments were recorded on a Bruker Avance III 400 spectrometer equipped with an 89 mm wide-bore 9.4 T superconducting magnet yielding Larmor Frequencies of 54.2 and 400 MHz, respectively. All of the samples were packed into 3.2 mm rotors inside a $N_2$-filled glove box. $^{17}$O and $^{1}$H chemical shifts are referenced to $H_2O$ at 0.0 ppm and to adamantane at 1.92 ppm, respectively.

$^{17}$O direct dynamic nuclear polarization (DNP) NMR experiments were performed at a Larmor frequency of 81.3 MHz on a 14.1 T Bruker Avance III HD 600 spectrometer equipped with a 395 GHz gyrotron microwave source and a 3.2 mm MAS probe. The microwave source power applied for $^{17}$O direct DNP

measurements was 7.0 W. $^{17}$O labeled ceria nanocubes were mixed with radical solution (16 mM TEKPol[44] in dried tetrachloroethane, TCE) in an Ar-filled glove box. $^{17}$O chemical shifts of the DNP NMR spectra were referenced to bulk ceria at 875 ppm at 100 K.

**Details of DFT calculations.** All spin-polarized DFT calculations were carried out using the *Vienna Ab initio Simulation Package* (VASP)[48]. The Perdew-Burke-Ernzerhof (PBE) functional[49] with the Hubbard U correction (DFT + U)[50] were used for all calculations. The effective U value of 5.0 eV was only applied to the localized Ce 4f orbitals[51,52]; our previous study shows that the calculated chemical shifts from PBE + U (5.0 eV) are in quantitative agreement with the experimental values[28]. The projector augmented wave method[53] was used to describe the interaction between core and valence electrons. A plane-wave kinetic energy cutoff of 500 eV was used for all calculations. For geometry optimization, all of the atoms were allowed to relax until the Hellman–Feynman forces were lower than 0.02 eV Å$^{-1}$. For electronic minimization, we used an energy convergence criterion of $10^{-5}$ eV for optimizing geometries and a higher criterion of $10^{-8}$ eV for chemical shift and electric field gradients (EFGs) calculations[28]. The optimized lattice parameter of ceria using PBE + U (5.0 eV) is 5.448 Å, which is in reasonable agreement with the experimental value (5.411 Å)[54].

We used a 2 × 2 surface cell to model the ceria (100) surface. The ceria (100) surface slab model with 12 oxygen layers was found to be sufficiently thick, i.e., the middle layers of this model mimic the bulk environment in terms of chemical shift (Supplementary Figs. 9, 12, 15, 18, 22, 30, 33, 36 and 38). All the slabs contain a large vacuum gap (>10 Å) to remove the slab-slab interactions. The k-point mesh was sampled by using a 2 × 2 × 1 Monkhorst–Pack grid.

We used the same method as our previous work[28] to calculate chemical shifts, quadrupole coupling constants ($C_Q$) and asymmetry parameters ($\eta$). For the electric EFG calculations to obtain $C_Q$ and $\eta$ of oxygen species, we used the experimental quadrupole moment (Q) of $-0.02558$ barns[55] for $^{17}$O. For calculating the isotropic chemical shift ($\delta_{iso}$), we used the following equation:

$$\delta_{iso} = \delta_{cal} + \delta_{ref}, \quad (1)$$

where $\delta_{cal}$ is the unaligned DFT chemical shift, $\delta_{ref}$ is the reference chemical shift. The averaged value of the unaligned DFT chemical shifts of oxygen species in the middle layers (layers 4–9) of every prototype slab models is 835 ppm. By aligning 835 ppm to the corresponding experimental value of 877 ppm, we obtained the $\delta_{ref}$ of 42 ppm.

The average adsorption energies of each water molecule ($E_{ads}$) on the (100) surface with the O-t or $CeO_4$-t model were calculated as the following:

$$E_{ads} = 1/n \cdot \{E[nH_2O/CeO_2] - E[CeO_2] - nE[H_2O]\}, \quad (2)$$

where $n$ is the number of adsorbed water molecules, $E[nH_2O/CeO_2]$, $E[CeO_2]$ and $E[H_2O]$ are the DFT calculated total energies of the adsorption complex, the ceria substrate and the gas phase $H_2O$ molecule, respectively.

Thermodynamic stabilities of different water adsorption structures on O-t and $CeO_4$-t surface units at given water partial pressure and temperature were determined by calculating the surface free energy per unit area ($\gamma(p,T)$)[41,56,57]:

$$\gamma(p, T) = 1/A \cdot \{G_{slab}[nH_2O/CeO_2](p, T) - mG_{bulk}[CeO_2](p, T) - n\mu[H_2O](p, T)\}, \quad (3)$$

where $A$ is the surface area of the slab, $n$ is the number of adsorbed water molecules, $m$ is the number of $CeO_2$ bulk (i.e., $Ce_4O_8$) units in the slab model, $\mu$ is the chemical potential, and $G$ is the Gibbs free energy.

We assumed that the surfaces are in thermodynamic equilibrium with gas phase $H_2O$. So, $\mu[H_2O]$ $(p,T)$ can be calculated as follows:

$$\mu[H_2O](p, T) = E[H_2O] + \Delta\mu[H_2O](p, T) = E[H_2O] + H[H_2O](p^0, T) - H[H_2O](p^0, 0K) - TS[H_2O](p^0, T) + K_B T ln(p/p^0), \quad (4)$$

where $p^0$ is the standard state pressure (0.1 MPa); enthalpy ($H$) and entropy($S$) terms were taken from the website of NIST[58]. As the DFT total energies of the solid components can be regarded as good approximations of corresponding Gibbs free energies[57], we then obtained:

$$\gamma(p, T) = 1/A \cdot \{G_{slab}[nH_2O/CeO_2](p, T) - mG_{bulk}[CeO_2](p, T) - nE[H_2O] - nH[H_2O](p^0, T) + nH[H_2O](p^0, 0K) + nTS[H_2O](p^0, T) - nK_B T ln(p/p^0)\}, \quad (5)$$

Note that the vibration contributions and the $pV$ ($V$ denotes volume) term of solid components were not considered.

## Data availability

The data that support the findings of this study are available from the corresponding author upon reasonable request.

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

## Acknowledgements

This work was supported by the National Natural Science Foundation of China (NSFC)−Royal Society Joint Program (21661130149) and NSFC (91745202, 21573103, and 21825301). The ECUST group also thanks the Programme of Introducing Talents of Discipline to Universities (B16017) and National Super Computing Centre in Jinan for computing time. L.P. thanks the Royal Society and the Newton Fund for Royal Society - Newton Advanced Fellowship. This work was also supported by a Project Funded by the Priority Academic Program Development of Jiangsu Higher Education Institutions. M.A. H. would like to thank the Oppenheimer foundation for funding. D.M.H. acknowledges the Cambridge International Trust for funding, and is grateful for support from NEC-CES, an Energy Frontier Research Center funded by the U.S. Department of Energy, Office of Science, Office of Basic Energy Sciences under Award No. DE-SC0012583. We

would like to thank Prof. Bingwen Hu. and Dr. Ming Shen in East China Normal University for invaluable discussions and help in this work.

## Author contributions

J.C. and K.Q. carried out the synthesis of ceria nanocubes; J.C., L.S., Y.W. and J.-H.D. carried out XRD, HRTEM, Raman, ICP-MS, N element analyzing, XPS and surface area measurement; J.C., T.L., M.A.H., P.C.M.M.M., L.S., Y.L., X.K. and L.P. performed $^{17}O$ isotope enrichment, and collected, as well as analyzed the $^{17}O$ and $^{1}H$ NMR spectra; J.C., M.A.H., D.M.H. and S.P. collected and analyzed the $^{17}O$ DNP NMR spectra; X.-P.W. and X.-Q.G. conducted the DFT calculations; J.C., X.-P.W., M.A.H., W.D., C.P.G. and L.P. wrote the manuscript, and all authors discussed the experiments and final manuscript.

## Competing interests

The authors declare no competing interests.
