## [Peer Review File · Nature Communications]

Reviewers' Comments:

Reviewer #1:

Remarks to the Author:

This contribution by Chen et al. is highly interesting and should be of broad audience for materials scientists involved in nanochemistry and advanced characterization techniques (including NMR/DNP for surface and subsequent bulk characterization).

Here, a nice combination of $^{17}\text{O}/^1\text{H}$ solid state NMR, DNP SENS and DFT calculations are proposed to the reader. The paper is clear, well written. Conclusions are convincing from my point of view, though some minor corrections have to be added. This contribution reaches the scientific standards of Nature Commun. and can be published after minor corrections.

My comments are the following:

- (1) lines 29,30: the limitations of EM have to be commented more. Currently: unclear to me.
- (2) line 49: vacuum conditions: the authors have to mention strong progress in environmental EM.
- (3) ref 16-18: not broad enough to illustrate the advantages of solid state NMR. More papers from other groups have to be mentioned in the bibliographic section.
- (4) caption of Figure 1: what are the interactions refocused by a "quadrupolar Hahn echo"? Please add some relevant references related to this sequence.
Insert in blue: please add some LB to help the reader to estimate the presence of smaller but important/relevant peaks.
- (5) line 119: ^1H chemical shifts are often related to the isolated/H-bonded character of the (hydroxyl- as an example) groups. The text has to be strengthened towards this particular direction (add pertinent references as well).
- (6) Figure 2, $^1\text{H}/^{17}\text{O}$ HETCOR experiment: did the authors include any homonuclear decoupling into the sequence to improve the indirect dimension resolution. If not, the authors should perform the experiment again.
Moreover, an advice could be to add an extra MQ MAS block (see Amoureux et al.) in order to enhance the ^{17}O resolution as well.
- (7) lines 148, 149: the sentence has to be (rather strongly) modulated. OK for: ^{17}O being " ..a more sensitive probe of the acidity..." by under MQ MAS conditions at ultra high magnetic field! And also by performing multi-field experiments (most probably).
- (8) line 193: the authors claim that "1162 and 1168 calculated values..." are not observed experimentally: OK.. but what are the "error bars" on such calculations... having in mind that all DFT calculations are performed at 0K and that motional processes are evoked in the paper! (ie line 234). This particular point has to be carefully clarified as it is key for a safe assignment procedure.
- (9) 1D ^1H spectra for quantitative measurements. Echoes are used: are they "quantitative" here? Did the authors try to improve the ^1H spectral resolution by using homonuclear decoupling to, at least, extract isotropic lines and then go back (in a second step) to more quantitative echo-MAS experiments? This could give more insight in the origin of the ^1H linewidths (I agree that it is actually a very difficult problem to tackle...)
- (10) Did the authors check carefully the eventual "probe /rotor/cap... contributions" in the ^1H spectra? Did they try to increment the number of rotor periods in the echo sequence?
- (11) line 234, again: what kind of dynamics do the authors have in mind? Unclear to me. To be illustrated.
- (12) the DNP section is rather impressive. However, I ask myself the following question: what is the influence of the DNP juice (to be explicitly mentioned by the way) on the surface state of the nanoparticles?

Reviewer #2:

Remarks to the Author:

Re: NCOMMS-19-07092

Multistep scenario in polarity compensation at oxide surfaces
by C. Yang et al.

The work presents an experimental and theoretical study of the atomic structure of the polar (100) surface of ceria. The topic is extremely interesting not only because it deals with one of the surfaces of ceria, which is a material that is of relevance in a number of technologically relevant applications such as catalysis, but also is of great relevance for our understanding of polarity compensation mechanisms at oxide surfaces in general.

Many works have addressed the nature of the termination of the CeO₂(100) surface and the existence of two nearly equally stable terminations have been proposed, namely, a so-called checkerboard arrangement of the surface oxygen atoms (O-t) and a CeO₄-termination (CeO₄-t).

It is the first time that ¹⁷O and ¹H solid-state NMR is used to characterize this surface, in combination with DFT calculations, and the results suggest that both CeO₄-pyramids and hydroxyls are present in the stable termination. This is potentially a very important finding, particularly because most interpretations of experimental results are based on checkerboard-type models. Hence, there is still room for surprises and many interpretations might need to be reconsidered.

The work is certainly of interest for the readership of Nature Communications but before I can make a definite recommendation regarding publication, I ask the authors to consider the following comments.

Though my field of expertise is not that of solid-state NMR, I have some comments/questions to the experimental part.

(i) The work uses ceria nanocubes as ideal models of the ceria(100) surface. It is said the HRTEM images show little evidence for (111) facets. In Supplementary Figure 2D, one observes some truncation of the corners. I wonder if one would be able to quantify the percentage of the (111) facets. Also, would not the shape of the nanocubes depend on temperature? A comment on these issues is desirable.

(ii) In Figures 1A and 1B, the ¹⁷O NMR spectrum for the NCs-¹⁷O₂ cubes and that for the NCs-H₂¹⁷O ones are shown, respectively. It is said that the peak at ~250 ppm is related to ¹⁷O-H groups. In the case of the NCs-¹⁷O₂ cubes, such a peak is practically non-existent (Figure 1A). Yet, in Figure 2, where ¹H NMR spectra are shown, the presence of H is observed for both samples, and the spectra for both cube types do not look too different. In the case of the NCs-¹⁷O₂, one imagines that one starts with NCs formed from ¹⁶O, and that some surface hydroxyls exist, i.e. ¹⁶O-H. Then, the cubes are ¹⁷O-labelled. After the ¹⁷O-labelling, H is observed (Figure 2A), but not the signal of the ¹⁷O that would be "below" that H (Figure 1A). Hence, one asks, why the isotopic exchange does not apparently occur on the ¹⁶O initially forming the ¹⁶O-H groups? A comment on this is expected.

(iii) It would be desirable to show IR spectra of these surfaces in the O-H vibrational region. Such spectra may help find additional fingerprints of the different hydroxyls on the surfaces.

I have some comments/questions to the modelling part.

(iv) In the DFT calculations, both O-t and CeO₄-t models were considered with 0, 1, 2, 3, and 4 water molecules. Having all that data, it would be relevant to indicate the relative stabilities of all these surfaces as a function of the chemical potential of water. Hence, constructing a phase diagram should not be complicated, and it will show important information (see e.g., Applied Surface Science.478, 68-74 (2019)).

(v) On the (111) surface, H₂O adsorbs molecularly and as a OH-H hydroxyl pair with very similar adsorption energy, but does not truly dissociate. In the case of the (100) terminations, I understand that water dissociates most probably without barrier.

What is the adsorption energy for molecularly chemisorbed water on the (100) terminations? There is much in the literature about the interaction of water with the low-index ceria surfaces, including the (100), and, at least in the Supplemental Material, an attempt should be made to compare with published data, particularly those obtained from similar DFT calculations (see e.g., J. Phys. Chem. C 2017, 121, 39, 21571-21578).

Reviewer #1 (Remarks to the Author):

This contribution by Chen et al. is highly interesting and should be of broad audience for materials scientists involved in nanochemistry and advanced characterization techniques (including NMR/DNP for source and subsequent bulk characterization). Here, a nice combination of $^{17}\text{O}/^1\text{H}$ solid state NMR, DNP SENS and DFT calculations are proposed to the reader. The paper is clear, well written. Conclusions are convincing from my point of view, though some minor corrections have to be added. This contribution reaches the scientific standards of Nature Commun. and can be published after minor corrections.

We thank the reviewer for the positive comments, efforts and time spent on refereeing our paper.

(1) lines 29,30: the limitations of EM have to be commented more. Currently: unclear to me.

Response: We thank the reviewer for the suggestion. Related discussion has now been modified in the abstract (paragraph 1, page 2). Now it reads: “The widely used electron microscopy techniques are limited because the volumes sampled may not be representative, and they provide little chemical bonding information with low contrast of light elements.”

In addition, related discussion has also been modified in the introduction part (paragraph 1, page 3): “Despite the many advantages of microscopy techniques, they are limited to the visualization of a small fraction of the sample which may not yield reliable quantitative information about the whole sample, and are typically performed at high vacuum conditions that may alter the surface environment.¹⁴ Furthermore, light elements, such as hydrogen and oxygen which are of key importance for many materials, are difficult to probe with such techniques.¹⁵”

14. Browning, N. D. *et al.* in *Modeling Nanoscale Imaging in Electron Microscopy* (eds. Vogt, T., Dahmen, W., & Binev, P.) 11–40 (Springer, 2012).
15. Senga, R. & Suenaga, K. Single-atom electron energy loss spectroscopy of light elements. *Nat. Commun.* **6**, 7943 (2015).

(2) line 49: vacuum conditions: the authors have to mention strong progress in environmental EM.

Response: We thank the reviewer for the helpful suggestion. Related discussion has been added in the revised manuscript and three references have been added (paragraph 1, page 3) : “Although significant developments have been made in environmental electron microscopy, which allows materials to be investigated under adjustable pressure conditions and in variable gaseous environments,^{16–18} the other disadvantages remain and complementary methods are required.”

16. Su, D. S., Zhang, B. & Schlögl, R. Electron microscopy of solid catalysts transforming from a challenge to a toolbox. *Chem. Rev.* **115**, 2818–288 (2015).
17. Jinschek, J. R. & Helveg, S. Image resolution and sensitivity in an environmental transmission electron microscope. *Micron* **43**, 1156–1168 (2012).
18. Jinschek, J. R. Atomic scale structure-function relationship of heterogeneous catalysts: Investigation of gas-solid interactions by ETEM. *Microsc. Anal.* **26**, S5–S10 (2012).

(3) ref. 16-18: not broad enough to illustrate the advantages of solid state NMR. More papers from other groups have to be mentioned in the bibliographic section.

Response: We thank the reviewer for this suggestion. Now more papers are cited to illustrate the advantages of solid-state NMR (paragraph 2, page 3).

19. Marchetti, A. *et al.* Understanding surface and interfacial chemistry in functional nanomaterials via solid-state NMR. *Adv. Mater.* **29**, 1605895 (2017).
20. Grey, C. P. & Dupré, N. NMR studies of cathode materials for lithium-ion rechargeable batteries. *Chem. Rev.* **104**, 4493–4512 (2004).
21. Zheng, A., Li, S., Liu, S. -B. & Deng, F. Acidic properties and structure-activity correlations of solid acid Catalysts revealed by solid-state NMR spectroscopy. *Acc. Chem. Res.* **49**, 655–663 (2016).
22. Bonhomme, C. *et al.* Advanced solid state NMR techniques for the characterization of sol-gel derived materials. *Acc. Chem. Res.* **40**, 738–746 (2007).
23. Salager, E. *et al.* Powder crystallography by combined crystal structure prediction and high-resolution ^1H solid-state NMR spectroscopy. *J. Am. Chem. Soc.* **132**, 2564–2566 (2010).
24. Deschamps, M. *et al.* Exploring electrolyte organization in supercapacitor electrodes with solid-state NMR. *Nat. Mater.* **12**, 351–358 (2013).
25. Kong, X. *et al.* Mapping of functional groups in metal-organic frameworks. *Science.* **341**, 882–885 (2013).
26. Peng, Y.-K. *et al.* Trimethylphosphine-assisted surface fingerprinting of metal oxide nanoparticle by ^{31}P solid-state NMR: A zinc oxide case study. *J. Am. Chem. Soc.* **138**, 2225–2234 (2016).
27. Salassa, G. & Bürgi, T. NMR spectroscopy: a potent tool for studying monolayer-protected metal nanoclusters. *Nanoscale Horiz.* **3**, 457–463 (2018).

(4) caption of Figure 1: what are the interactions refocused by a "quadrupolar Hahn echo"? Please add some relevant references related to this sequence. Insert in blue: please add some LB to help the reader to estimate the presence of smaller but important/relevant peaks.

Response: We thank the reviewer for pointing out this mistake and this helpful suggestion. We used a Hahn-echo pulse sequence ($\pi/6$ - τ - $\pi/3$ - τ -acquisition), and $\pi/6$ and $\pi/3$ pulses were used due to the quadrupolar nature of ^{17}O . We have removed the word "quadrupolar" in the caption of Figure 1 in the revised manuscript.

In order to show the peaks due to OH better, a line broadening of 600 Hz is applied for the enlarged spectrum shown in light blue region (430 ~ 45 ppm, Figure 1A, page 6; Figure 4A, page 13).

(5) line 119: ^1H chemical shifts are often related to the isolated/H-bonded character of the (hydroxyl- as an example) groups. The text has to be strengthened towards this particular direction (add pertinent references as well).

Response: We thank the reviewer for this helpful suggestions. Related discussion has been added in the revised manuscript (paragraph 2, page 7) with relevant references cited: " ^1H chemical shifts are sensitive to hydrogen bonding^{35,36} and the lineshapes can be affected by the distribution of hydrogen bond distances; in particular, stronger hydrogen bonding results ^1H NMR signals in higher frequency.³⁷"

35. Ratcliffe, C. I., Ripmeester, J. A. & Tse, J. S. NMR chemical shifts of dilute ^1H in inorganic solids. *Chem. Phys. Lett.* **120**, 427–432 (1985).
36. Gill, L. *et al.* Fast MAS ^1H NMR study of water adsorption and dissociation on the (100) surface of ceria nanocubes: A fully hydroxylated, hydrophobic ceria surface. *J. Phys. Chem. C* **121**, 7450–7465 (2017).
37. Kim, G., Blanc, F., Hu, Y. Y. & Grey, C. P. Understanding the conduction mechanism of the protonic conductor CsH_2PO_4 by solid-state NMR spectroscopy. *J. Phys. Chem. C* **117**, 6504–6515 (2013).

(6) Figure 2, $^1\text{H}/^{17}\text{O}$ HETCOR experiment: did the authors include any homo-nuclear decoupling into the

sequence to improve the indirect dimension resolution. If not, the authors should perform the experiment again. Moreover, an advice could be to add an extra MQ MAS block (see Amoureux et al.) in order to enhance the ^{17}O resolution as well.

Response: We thank the reviewer for the suggestions. 1D ^1H NMR experiments for $\text{NCs-}^{17}\text{O}_2$ and $\text{NCs-H}_2^{17}\text{O}$ with and without homonuclear decoupling were performed, and no obvious resolution improvement was observed (Figure X1). This data suggests that the linewidths of ^1H NMR signals of $\text{NCs-}^{17}\text{O}_2$ and $\text{NCs-H}_2^{17}\text{O}$ should arise mostly from the chemical shift distribution instead of $^1\text{H-}^1\text{H}$ homonuclear dipolar coupling. Therefore, a HETCOR experiment with ^1H homonuclear dipolar decoupling applied is not attempted.

Figure X1. ^1H MAS NMR spectra recorded with and without homonuclear decoupling for (A) $\text{NCs-}^{17}\text{O}_2$ and (B) $\text{NCs-H}_2^{17}\text{O}$. The “Without” spectra are measured with a Hahn-echo pulse sequence ($\pi/2$ - τ - π - τ -acquisition) where τ equals one rotor period at a MAS frequency of 8 kHz with a recycle delay of 4.0 s. The scaled “With” spectra are acquired by a $w\text{PMLG5}_p^{X\bar{X}}$ pulse sequence^{X1} with each pulse lasting 1.5 μs , utilizing the same spinning speed and recycle delay. The RF field strength for the $w\text{PMLG5}_p^{X\bar{X}}$ is 100 kHz. These experiments are recorded on a Bruker Avance III 400 spectrometer equipped with a 4.0 mm probe.

Adding an MQMAS block in the HETCOR experiment will probably enhance the resolution in ^{17}O dimension, generating a spectrum free of broadening from 2nd order quadrupolar effects, however, this would require a prohibitively long experimental time, due to the small amount of surface species and the reduction of signal intensities because of the low sensitivity of MQMAS^{X2} (It already took more than one day to acquire the 2D spectrum in Figure 2C). Furthermore, resolving different hydroxyl sites are beyond the scope of the current paper, so we have not tried to do the MQ-HETCOR experiments.

But we did try to resolve different ^{17}O (OH) sites according to the contour plot and ^1H dimension of $^1\text{H} \rightarrow ^{17}\text{O}$ HETCOR spectrum (Figure 2C). Although the S/N is not great, four ^{17}O NMR slices were extracted and they were fitted by using the Dmfit program.^{X3} The results are shown in Figure X2 and Table X1. The NMR parameters extracted for the four hydroxyl sites are quite close to the calculated data, indicating that it is possible to fully resolve these different species at a higher magnetic field.

Figure X2. $^1\text{H} \rightarrow ^{17}\text{O}$ HETCOR NMR spectrum of $\text{NCS-H}_2^{17}\text{O}$ (A) with the extracted ^{17}O NMR slices and corresponding NMR simulations (B – E). The ^{17}O NMR parameters used in the simulations, including isotropic chemical shifts (δ_{iso}), quadrupolar parameters (C_Q and η) and center of gravity (δ_{CG}) are summarized in Table X1.

Table X1. The comparison of ^{17}O NMR parameters simulated based on the experimental data (Figure X2) and calculated with DFT.

	Extracted in B		Extracted in C		Extracted in D		Extracted in E	
	Calculated ^a	Simulated	Calculated ^a	Simulated	Calculated ^a	Simulated	Calculated ^a	Simulated
$\delta_{\text{iso}} / \text{ppm}$	286	286	326	326	343	343	385	385
$C_{\text{Q}} / \text{MHz}$	6.23	6.23	6.85	6.65	6.78	6.18	6.74	6.24
η	0.15	0.54	0.12	0.58	0.08	0.58	0.19	0.58
$\delta_{\text{CG}} / \text{ppm}$	206	200	230	226	249	239	291	297

^aThe DFT calculations are based on the model M_3 (see Supplementary Figure 37 for the structure) which can represent the surface structure of $\text{NCs-H}_2^{17}\text{O}$. The calculated ^{17}O parameters are summarized in Supplementary Table 13 in the revised SI file.

- X1. Leskes, M., Madhu, P. K. & Vega, S. A broad-banded z -rotation windowed phase-modulated Lee–Goldburg pulse sequence for ^1H spectroscopy in solid-state NMR. *Chem. Phys. Lett.* **447**, 370–374 (2007).
- X2. Ashbrook, S. E., Wimperis, S. High-resolution NMR of quadrupolar nuclei in solids: the satellite-transition magic angle spinning (STMAS) experiment. *Prog. Nucl. Magn. Reson. Spectrosc.* **45**, 53–108 (2004).
- X3. Massiot, D. *et al.* Modelling one- and two-dimensional solid-state NMR spectra. *Magn. Reson. Chem.*, **40**, 70–76 (2002).

(7) lines 148, 149: the sentence has to be (rather strongly) modulated. OK for: ^{17}O being " ..a more sensitive probe of the acidity..." by under MQ MAS conditions at ultra high magnetic field! And also by performing multi-field experiments (most probably).

Response: We thank the reviewer for the helpful suggestion and comment. Related discussion has been modified in the manuscript (paragraph 2, page 8): "Since the ^{17}O NMR shift range is much wider than that for ^1H NMR shifts, ^{17}O NMR spectroscopy may provide an alternative and more sensitive probe of the acidity and acid-catalysis reactivity for oxide nanomaterials. However, ^{17}O NMR experiments are often required to perform at multiple magnetic fields, preferentially higher fields, or with high resolution techniques (e.g., MQMAS⁴⁰), in order to decrease the linewidths arising from quadrupolar interactions."

40. Medek, A., Harwood, J. S. & Frydman, L. Multiple-quantum magic-angle spinning NMR: A new method for the study of quadrupolar nuclei in solids. *J. Am. Chem. Soc.* **117**, 12779–12787 (1995).

(8) line 193: the authors claim that "1162 and 1168 calculated values..." are not observed experimentally: OK.. but what are the "error bars" on such calculations... having in mind that all DFT calculations are performed at 0 K and that motional processes are evoked in the paper! (ie line 234). This particular point has to be carefully clarified as it is key for a safe assignment procedure.

Response: We thank the reviewer for the comments. Although our DFT calculations were performed at 0 K, our previous ^{17}O solid-state NMR study on ceria (111)^{X4} demonstrates that the calculated ^{17}O chemical shifts using the Perdew-Burke-Ernzerhof (PBE) functional with the Hubbard U correction (PBE + U) are in good agreement with the experimental results; the differences of the calculated (0 K) and experimental results (room temperature) are generally around or less than 10 ppm, suggesting that the temperature effect is trivial. We have now mentioned this point explicitly in the main text. Considering the very large frequency differences of different characteristic signals in the experimental ^{17}O NMR spectra (e.g. the resonances at 935, 970 and ~1012 ppm, see Figure 1 in the revised manuscript), we believe that the spectral assignments are reliable.

The calculated chemical shifts of 1162 and 1168 ppm are associated with bare surface $\text{O}_{2\text{C}}$ ions, which is consistent with the conclusion from our previous study that ^{17}O NMR shift of oxygen ions in ceria increases with decreasing coordination number.^{X4} DFT calculations (Supplementary Figures 25 – 29 in the revised SI) suggest that bare $\text{O}_{2\text{C}}$ species react with water to form hydroxyl groups, which are related to peaks with more negative shifts. The absence of the ^{17}O NMR signals at 1162 and 1168 ppm suggests the complete conversion of surface $\text{O}_{2\text{C}}$ ions to hydroxyl groups, rather than the change of spectra due to motional processes. These results are in agreement with the presence of the small ^1H NMR signals at 5.4 ppm due to molecular water in the experimental ^1H NMR spectra (Figure 2 in the revised manuscript).

Related discussion has been added to the revised manuscript (paragraph 1, page 10): "The differences of the calculated and experimental results are generally around or less than 10 ppm,²⁸ allowing reliable spectral assignment."

X4. Wang, M. *et al.* Identification of different oxygen species in oxide nanostructures with ^{17}O solid-state NMR spectroscopy. *Sci. Adv.* **1**, e1400133 (2015).

28. Wang, M. *et al.* Identification of different oxygen species in oxide nanostructures with ^{17}O solid-state NMR spectroscopy. *Sci. Adv.* **1**, e1400133 (2015).

(9) 1D ^1H spectra for quantitative measurements. Echoes are used: are they "quantitative" here? Did the authors try to improve the ^1H spectral resolution by using homonuclear decoupling to, at least, extract isotropic

lines and then go back (in a second step) to more quantitative echo-MAS experiments? This could give more insight in the origin of the ^1H linewidths (I agree that it is actually a very difficult problem to tackle...)

Response: We thank the reviewer for this important remark. We performed the experiments on the samples and the references again with single pulse and Hahn echo sequences, and confirmed that Hahn echo sequences can be used.

Here Figure X3 shows that the results obtained with two pulse sequences are very similar.

Figure X3. ^1H NMR spectra of (A) NCs- $^{17}\text{O}_2$ and (B) NCs- H_2^{17}O obtained by using single pulse and Hahn echo sequence. The background signals (empty rotor including cap, rotor and probe etc.) of these spectra are subtracted (also see the following response to reviewers). Spinning speed: 12 kHz; recycle delay: 2.0 s.

As discussed in response to reviewer's comment #6, the ^1H signals are relatively broad mostly because of chemical shift distribution rather than homonuclear decoupling.

(10) Did the authors check carefully the eventual "probe /rotor/cap... contributions" in the ^1H spectra? Did they try to increment the number of rotor periods in the echo sequence?

Response: We thank the reviewer for the important remark. In our original manuscript, the contributions from the probe, rotor, cap etc. have been taken care of by subtracting the spectrum of an empty rotor obtained at same condition from the initial spectrum obtained. We have now added a statement in the caption of Figure 2 in the revised manuscript to specify how it is done.

However, we did not try to increase the number of rotor periods in the echo sequence initially. We have now performed those experiments on the samples and the results are shown in Figures X4 and X5.

Figure X4. The raw, background and subtracted ^1H NMR spectra of $\text{NCs-}^{17}\text{O}_2$ and $\text{NCs-H}_2^{17}\text{O}$ recorded with single pulse sequence and Hahn-echo sequence with different time between the $\pi/2$ and π pulses. Spinning speed: 12 kHz; recycle delay: 2.0 s.

Increasing the number of rotor periods results in only very slight differences in the lineshape and relative intensity (Figure X5).

Figure X5. ^1H NMR spectra of (A) NCs- $^{17}\text{O}_2$ and (B) NCs- H_2^{17}O acquired with Hahn echo pulse sequences with 1 – 3 rotor periods between the $\pi/2$ and π pulses. The background signals have been removed (see Figure X4 (C – H)). Spinning speed: 12 kHz; recycle delay: 2.0 s.

(11) line 234, again: what kind of dynamics do the authors have in mind? Unclear to me. To be illustrated.

Response: The environment giving rise to the calculated shift of 756 ppm, which is not experimentally observed, is highly distorted. The zero-temperature DFT calculations find one low energy local minimum with this distortion, however it is likely that there are many distortions of similar energy (and which may indeed be equivalent by symmetry), so that at ambient temperatures there is rapid interconversion of these local minima and thus dynamic averaging out of the distortion (i.e. dynamics of the oxygen ions). Alternatively, the distortion may be lost on the formation of a hydroxyl group between the CeO_4 reconstructions, as also suggested by the calculations. We have now modified the main text (paragraph 1, page 14) to clarify this discussion: “This distortion may be lost on addition of a bridging hydroxyl between the CeO_4 reconstructions (e.g. H_4 , see Figure 6), which is above the distorted environment, or the distortion may be averaged at non-zero temperatures due to rapid interconversion of different local environments with similar energies.”

(12) The DNP section is rather impressive. However, I ask myself the following question: what is the influence of the DNP juice (to be explicitly mentioned by the way) on the surface state of the nanoparticles?

Response: We thank the reviewer for the positive recognition on the DNP section. According to previously published results,^{X5} the biradical prefers to lay parallel on the sample surface, the favored configuration has an upright orientation with its -NO group interacting with the surface hydroxyls via hydrogen bonding. The solvent, TCE, is non-polar and is expected to have minimal interactions with the surface. The influence of DNP juice on the oxygen ions at the ceria (100) surfaces is thus expected to be minimal, as previously shown on ceria (111) surfaces.^{X6}

X5. Perras, F. A. *et al.* Optimal sample formulations for DNP SENS: The importance of radical-surface interactions. *Curr. Opin. Colloid Interface Sci.* **33**, 9–18 (2018).

X6. Hope, M. A. *et al.* Surface-selective direct ^{17}O DNP NMR of CeO_2 nanoparticles. *Chem. Commun.* **53**, 2142–2145 (2017).

Reviewer #2 (Remarks to the Author):

Re: NCOMMS-19-07092 *Multistep scenario in polarity compensation at oxide surfaces by C. Yang et al.* The work presents an experimental and theoretical study of the atomic structure of the polar (100) surface of ceria. The topic is extremely interesting not only because it deals with one of the surfaces of ceria, which is a material that is of relevance in a number of technologically relevant applications such as catalysis, but also is of great relevance for our understanding of polarity compensation mechanisms at oxide surfaces in general. Many works have addressed the nature of the termination of the $\text{CeO}_2(100)$ surface and the existence of two nearly equally stable terminations have been proposed, namely, a so-called checkerboard arrangement of the surface oxygen atoms (O-t) and a CeO_4 -termination ($\text{CeO}_4\text{-t}$). It is the first time that ^{17}O and ^1H solid-state NMR is used to characterize this surface, in combination with DFT calculations, and the results suggest that both CeO_4 -pyramids and hydroxyls are present in the stable termination. This is potentially a very important finding, particularly because most interpretations of experimental results are based on checkerboard-type models. Hence, there is still room for surprises and many interpretations might need to be reconsidered. The work is certainly of interest for the readership of *Nature Communications* but before I can make a definite recommendation regarding publication, I ask the authors to consider the following comments.

We thank the reviewer for the positive comments, efforts and time spent on refereeing our paper.

Though my field of expertise is not that of solid-state NMR, I have some comments/questions to the experimental part.

(i) The work uses ceria nanocubes as ideal models of the ceria(100) surface. It is said the HRTEM images show little evidence for (111) facets. In Supplementary Figure 2D, one observes some truncation of the corners. I wonder if one would be able to quantify the percentage of the (111) facets. Also, would not the shape of the nanocubes depend on temperature? A comment on these issues is desirable.

Response: We thank the reviewer for the comments. In Supplementary Figure 2D, truncations of the corners can be observed which may expose (111) surfaces. However, no characteristic peak at around 1030~1040 ppm corresponding to oxygen ions at 1st layer of ceria (111) surfaces^{X4} can be observed in the ^{17}O NMR spectrum of $\text{NCs-}^{17}\text{O}_2$ or $\text{NCs-H}_2^{17}\text{O}$ (Figure 1 in the revised manuscript), indicating that the fraction of truncations is quite small.

We believe that quantification of the percentages of facets based on the HRTEM images may be too subjective and such data may not be representative of the whole sample. At the same time, quantification according to ^{17}O NMR data will be problematic if ^{17}O exchange with oxygen ions on different facets differently (discussion at the end of page 3 in the revised manuscript).

The shapes of the nanocubes indeed depend on the temperature. The ceria nanocubes can present more obvious truncations at the corners when they are annealed above 673 K (in this work, the nanocubes were annealed at 573 K). Related investigations by applying NMR are on the way.

X4. Wang, M. *et al.* Identification of different oxygen species in oxide nanostructures with ^{17}O solid-state NMR spectroscopy. *Sci. Adv.* **1**, e1400133 (2015).

(ii) In Figures 1A and 1B, the ^{17}O NMR spectrum for the $\text{NCs-}^{17}\text{O}_2$ cubes and that for the $\text{NCs-H}_2^{17}\text{O}$ ones are shown, respectively. It is said that the peak at ~250 ppm is related to $^{17}\text{O-H}$ groups. In the case of the $\text{NCs-}^{17}\text{O}_2$ cubes, such a peak is practically non-existent (Figure 1A). Yet, in Figure 2, where ^1H NMR spectra are shown, the presence of H is observed for both samples, and the spectra for both cube types do not look too different. In the case of the $\text{NCs-}^{17}\text{O}_2$, one imagines that one starts with NCs formed from ^{16}O , and that some surface hydroxyls exist, i.e. $^{16}\text{O-H}$. Then, the cubes are ^{17}O -labelled. After the ^{17}O -labelling, H is observed (Figure 2A),

but not the signal of the ^{17}O that would be "below" that H (Figure 1A). Hence, one asks, why the isotopic exchange does not apparently occur on the ^{16}O initially forming the ^{16}O -H groups? A comment on this is expected.

Response: We thank the reviewer for the insightful comments. The isotopic exchange does occur on the OH group. We have now replotted Figure 1 with more line broadening, which shows the broad peaks of ^{17}O -H better. The peaks arising from ^{17}O -H groups in the ^{17}O spectrum of NCs- $^{17}\text{O}_2$ have a much lower intensity compared to the intense signal at 877 ppm arising from the oxygen ions in the bulk part of the nanorods. However, their intensities (^{17}O in OH is associated with large quadrupolar interaction) are similar to that of the relatively sharp resonances at 970 and 935 ppm ascribed to the oxygen ions at the 1st layer (small quadrupolar interaction). Therefore, both surface O and OH ions are labeled with ^{17}O similarly, and it is the different peak widths due to quadrupolar interactions that makes the latter peaks appear much "smaller" than the former.

(iii) It would be desirable to show IR spectra of these surfaces in the O-H vibrational region. Such spectra may help find additional fingerprints of the different hydroxyls on the surfaces.

Response: Figure X6 shows the bands centered at 1055, 952 and 853 cm^{-1} , which can be ascribed to the O_W -H twisting, O_S -Hs twisting and O_S -H- O_W rocking librations of surface hydroxyl groups,^{X7} where O_S and O_W denote oxygen ions at the ceria surface and in the water molecules, respectively. The frequencies observed for NCs- $^{17}\text{O}_2$ and NCs- H_2^{17}O are very similar, making it difficult to extract detailed surface structure information.

X7. Fernández-Torre, D. *et al.* Insight into the adsorption of water on the clean $\text{CeO}_2(111)$ surface with van der Waals and hybrid density functionals. *J. Phys. Chem. C* **116**, 13584–13593 (2012).

Figure X6. FT-IR spectra of the NCs- $^{17}\text{O}_2$ and NCs- H_2^{17}O .

I have some comments/questions to the modelling part.

(iv) In the DFT calculations, both O-t and CeO_4 -t models were considered with 0, 1, 2, 3, and 4 water molecules. Having all that data, it would be relevant to indicate the relative stabilities of all these surfaces as a function of the chemical potential of water. Hence, constructing a phase diagram should not be complicated, and it will show important information (see e.g., *Applied Surface Science*. 478, 68-74 (2019)).

Response: We thank the reviewer for this helpful suggestion. To better show the temperature and pressure effects that are incorporated in the chemical potential of water, relevant phase diagrams were constructed.

In the phase diagrams, the surface free energies of different structures as a function of temperature at four relevant water pressures (1, 10, 100, and 1000 Pa, see Supplementary Figure 39 in the revised SI file) were calculated. These four water pressures were chosen based on the annealing procedure (573 K in flowing dry O₂ gas) before ¹⁷O isotropic labeling.

We have added the following discussion in Supplementary Note 2 in the revised SI file (page 61): “Surface free energies of different structures including the O-t model with 0, 1, 2, 3, and 4 dissociative water molecules per surface unit and the CeO₄-t model with 0, 1, 2, and 3 dissociative water molecules per surface unit (M₀, M₁, M₂, and M₃) were calculated in Supplementary Figure 39 to show the relative stabilities of these structures at experimental conditions. Considering that surface reconstructions generally occur at harsh conditions,⁹ the ceria (100) surface structure should form during the annealing procedure before ¹⁷O isotropic labeling (at 573 K with a water pressure between 1 and 1000 Pa) and remain unchanged by the ¹⁷O isotropic labeling process and NMR measurements at relatively mild conditions (a temperature lower than 573 K and water pressure higher than 1000 Pa). Supplementary Figure 39 shows that M₁ is the most stable model at 573 K under water pressures ranging from 1 to 1000 Pa. This further suggests the occurrence of CeO₄ reconstructions on the ceria (100) surface before ¹⁷O enrichment. The following isotopic labeling process causes an increase of water coverage, resulting in the presence of corresponding surface structures such as M₂ and M₃ in NCs-¹⁷O₂ and NCs-H₂¹⁷O, respectively.”

9. Pan, Y. *et al.* Ceria nanocrystals exposing wide (100) facets: Structure and polarity compensation. *Adv. Mater. Interfaces* **1**, 1400404 (2014). (In the SI)

Supplementary Figure 39. Calculated surface free energies (γ) of different structures as a function of temperature (T) and water pressures (p) of (A) 1, (B) 10, (C) 100, and (D) 1000 Pa.

(v) On the (111) surface, H₂O adsorbs molecularly and as an OH-H hydroxyl pair with very similar adsorption energy, but does not truly dissociate. In the case of the (100) terminations, I understand that water dissociates most probably without barrier. What is the adsorption energy for molecularly chemisorbed water on the (100) terminations? There is much in the literature about the interaction of water with the low-index ceria surfaces, including the (100), and, at least in the Supplemental Material, an attempt should be made to compare with published data, particularly those obtained from similar DFT calculations (see e.g., *J. Phys. Chem. C* 2017, 121, 39, 21571-21578).

We thank the reviewer for the valuable suggestion. A previous computational study reported that dissociative adsorption of water is much more favorable (-1.65 eV) than that of molecular adsorption (-0.94 eV) on ceria (100) O-t surface.⁴¹ Therefore, the adsorption of molecular water on the ceria (100) O-t surface was not considered. We have added this sentence in the revised manuscript: “A previous computational study reported that dissociative adsorption of water is much more favorable than the molecular adsorption of water on the (100) O-t surface;⁴¹” (paragraph 1, page 11) Our calculations show that the adsorption energy (E_{ads}) for a dissociatively adsorbed water molecule on the (100) $p(2 \times 2)$ surface with the O-t model is -1.47 eV, while the adsorption energies for a molecularly / dissociatively adsorbed water on the (100) $p(2 \times 2)$ surface with the CeO₄-t model is -0.76/-1.86 eV (Supplementary Table 9). The average adsorption energies of each water molecule on the (100) $p(2 \times 2)$ surface with the O-t or CeO₄-t models at different water concentrations are summarized in Supplementary Table 9 in the revised SI file.

Supplementary Table 9. Calculated average adsorption energies of each water molecule (E_{ads} , in eV / H₂O) on the (100) O-t or CeO₄-t surfaces at molecular adsorption or dissociative adsorption (note that only the structure with the most negative energy is considered for each case), in comparison to the values reported previously.

Number of Water Molecules per surface unit		O-t Surface Unit		CeO ₄ -t Surface Unit	
		Molecular / eV	Dissociative / eV	Molecular / eV	Dissociative / eV
1	Our work	–	-1.47	-0.76	-1.86
	Ref 8.	-0.94 ^a	-1.65 ^a	–	–
2	Our work	–	-1.21	–	-1.47
	Ref 8.	–	-1.53 ^a	–	–
3	Our work	–	-1.38	–	-1.21
4	Our work	–	-1.31	–	–

^a PBE + *U* (4.5 eV for Ce 4f) + D2 value reported for a $p(2 \times 2)$ surface cell;⁸

41. Kropp, T., Paier, J. & Sauer, J. Interactions of water with the (111) and (100) surfaces of ceria. *J. Phys. Chem. C* **121**, 21571–21578 (2017).
8. Kropp, T., Paier, J. & Sauer, J. Interactions of water with the (111) and (100) surfaces of ceria. *J. Phys. Chem. C* **121**, 21571–21578 (2017). (In the SI)

References in the response to reviewer:

- X1. Leskes, M., Madhu, P. K. & Vega, S. A broad-banded z -rotation windowed phase-modulated Lee–Goldburg pulse sequence for ^1H spectroscopy in solid-state NMR. *Chem. Phys. Lett.* **447**, 370–374 (2007).
- X2. Ashbrook, S. E., Wimperis, S. High-resolution NMR of quadrupolar nuclei in solids: the satellite-transition magic angle spinning (STMAS) experiment. *Prog. Nucl. Magn. Reson. Spectrosc.* **45**, 53–108 (2004).
- X3. Massiot, D. *et al.* Modelling one- and two-dimensional solid-state NMR spectra. *Magn. Reson. Chem.*, **40**, 70–76 (2002).
- X4. Wang, M. *et al.* Identification of different oxygen species in oxide nanostructures with ^{17}O solid-state NMR spectroscopy. *Sci. Adv.* **1**, e1400133 (2015).
- X5. Perras, F. A. *et al.* Optimal sample formulations for DNP SENS: The importance of radical-surface interactions. *Curr. Opin. Colloid Interface Sci.* **33**, 9–18 (2018).
- X6. Hope, M. A. *et al.* Surface-selective direct ^{17}O DNP NMR of CeO_2 nanoparticles. *Chem. Commun.* **53**, 2142–2145 (2017).
- X7. Fernández-Torre, D. *et al.* Insight into the adsorption of water on the clean $\text{CeO}_2(111)$ surface with van der Waals and hybrid density functionals. *J. Phys. Chem. C* **116**, 13584–13593 (2012).

Reviewers' Comments:

Reviewer #1:

Remarks to the Author:

I (reviewer 1) have looked carefully at the revised version of this contribution by Peng et al. The revisions are overall positive and the authors have made a very nice work including new NMR control experiments. I feel that their intention has been to clarify their paper as much as possible in full intellectual honesty. I appreciate a lot! The bibliographic section has been extended as well which is a good point for the overall impact of the paper.

Moreover, I have looked at the corrections related to rev. 2 comments. Again, the authors have made a careful analysis of the raised questions and have proposed detailed answers.

From my point of view, the paper in its current version reaches now the scientific standards of Nature Commun and can be published as it is.

Reviewer #2:

Remarks to the Author:

Review of NCOMMS-19-07092A:

"Polar Surface Structure of Oxide Nanocrystals Revealed with Solid-State NMR Spectroscopy" by Chen et al.

The authors have taken all comments by all reviewers very seriously and have responded them by adding new discussions in the manuscript, as well as in the supplemental material. As for my questions and comments, I am quite pleased with the responses, which I think have improved the presentation of the work. I do recommend its publication in Nature Communications.